# Suppression of Mainlobe Jammers with Quadratic Element Pulse Coding in MIMO Radar

**Yiqun Zhang, Guisheng Liao, Lan Lan** **, Jingwei Xu \*** **and Xuepan Zhang**

National Key Laboratory of Radar Signal Processing, Xidian University, Xi'an 710071, China;
yiqzhang@stu.xidian.edu.cn (Y.Z.); liaogs@xidian.edu.cn (G.L.); lanlan@xidian.edu.cn (L.L.);
zhangxuepan1@xidian.edu.cn (X.Z.)
**\*** Correspondence: jwxu@xidian.edu.cn; Tel.: +86-181-9203-9179

**Abstract:** The problem of suppressing mainlobe deceptive jammers, which spoof radar systems by generating multiple false targets, has attracted widespread attention. To tackle this problem, in this paper, the multiple-input multiple-output (MIMO) radar system was utilized by applying a quadratic element phase code (QEPC) to the transmitted pulses of different elements. In the receiver, by utilizing the spatial frequency and Doppler frequency offset generated after decoding, the jammers were equivalently distributed in the sidelobes of the joint Doppler-transmit-receive domain and were distinguishable from the true target. Then, further spatial frequency compensation and Doppler compensation were performed to align the true target to the zero point in the transmit spatial and Doppler domains. Moreover, by designing appropriate coding coefficients, the jammers were suppressed by data-independent Doppler-transmit-receive three-dimensional beamforming. However, the beamforming performance was sensitive to angular estimation mismatches, resulting in performance degradation of jammer suppression. To this end, a center-boundary null-broadening control (CBNBC) approach was used to broaden the nulls in the equivalent beampattern by generating multiple artificial jammers with preset powers around the nulls. Thus, the false targets (FTs) with deviations were sufficiently suppressed in the broadened notches. Numerical simulations and theoretical analysis demonstrated the performance of the developed jammer suppression method.

**Keywords:** QEPC-MIMO radar; mainlobe deceptive jammer suppression; center-boundary null-broadening control

## 1. Introduction

Radar systems face severe challenges in complex electromagnetic environments, where jammers degrade their ability to detect targets [1,2]. Among the various types of jammers, deceptive jammers are quite threatening because they intercept and retransmit radar signals [3]. Particularly, a large number of false targets (FTs) can be generated through the digital radio frequency memory (DRFM) after modulation with an appropriate time delay, which makes the radar system track the FTs by mistake [4]. When the FTs are located in the sidelobe, spatial processing methods, including the generalized sidelobe canceller (GSC) [5], ultra-low sidelobe antennas [6], and space-time adaptive processing (STAP) [7,8], can be utilized to suppress the jamming signal. Nevertheless, the mainlobe deceptive jammers are indistinguishable from the true target in the angle domain; it is difficult to suppress mainlobe jammers with traditional spatial processing methods.

To tackle the mainlobe deceptive jammer suppression problem, methods have been explored in several domains, such as the frequency domain [9–13], spatial domain [14–16], time domain [17,18], and polarization domain [19], to find the differences between the FTs and the true one. In the frequency domain, using a compressed sensing framework, a suppression method to suppress the narrow-band jamming signal was investigated [20]. However, the method was limited to other types of jammers. Moreover, pulse frequency

agility was utilized to enhance the robustness of the system [21]. However, the coherence among pulses was destroyed by the different frequencies among the transmit pulses. In the spatial domain, a projection matrix was investigated to design data preprocessing matrices to overcome the mainlobe distortion and suppress the jammers. Nevertheless, considering the existence of deviations, it was difficult to construct the projection matrix precisely [22]. In the time domain, the desired echoes could be separated via the blind source separation algorithm [23]. However, the number of sources needed to be estimated in advance. In the polarization domain, the spatial polarization characteristics were utilized to suppress the jamming through polarization estimation and orthogonal polarization decomposition [24]. In fact, it is difficult to distinguish the true target from FTs with traditional radar systems. Hence, there is an urgent demand to explore suppression methods in novel radar systems.

During the past years, frequency diverse array-multiple-input multiple-output (FDA–MIMO) radar has attracted intense interest [25–29]. In the FDA configuration, a small frequency increment is introduced across the transmit array elements, offering additional degrees of freedom (DOFs) in the range domain [30–34]. Using adaptive beamforming, FTs are suppressed in the FDA-MIMO radar due to range mismatch [35–38]. Nevertheless, some problems exist. (1) In adaptive beamformers, if it is difficult to guarantee the independent and identically distributed (IID) condition, the performance of jammer suppression degrades. (2) Some array mismatches, such as the quantization error in both range and angle domains, reduce the performance of mainlobe jammer suppression. (3) The ability of jammer suppression is limited to the array configuration, namely, the suppression performance is no longer effective when the maximum number of jammers is larger than that of the transmit elements. In this regard, with an appropriate coding coefficient and preset beampattern synthesis, the mainlobe deceptive jammers can be suppressed in Element Pulse-Coding (EPC)-MIMO, through which the aforementioned problems (1) and (2) can be solved [39]. However, it is not able to tackle the problem (3). Hence, there is a need for effective approaches to increase the maximum number of suppressible jammers.

Following the guidelines, a novel coding scheme based on the MIMO configuration was developed. In the transmit array, the quadratic element phase code (QEPC) was devised by modulating the phases of pulses along the slow time dimension with a fixed quadratic coding coefficient in the distinct element. In this respect, additional DOFs in the joint Doppler and spatial frequency domain were obtained. In the receiver, the spatial frequency and Doppler frequency offset were generated after the decoding procedure. In this respect, the jammers were equivalently distributed in the sidelobes in the joint Doppler-transmit-receive domain. Hence, the FTs were distinguishable from the true target. Then, further spatial frequency compensation and Doppler compensation were performed to align the true target to the zero point in the transmit-spatial and Doppler domains. Moreover, by designing appropriate coding coefficients, the jammers were suppressed via the data-independent Doppler-transmit-receive three-dimensional beamformer. Considering that the actual FTs deviated from their presumed nulls and could not be adequately suppressed due to spatial frequency mismatch, a center-boundary null-broadening control (CBNBC) approach was used. It broadened the nulls by imposing artificial jammers from the center to the preset boundary in the equivalent beampattern with preset powers around the nulls. Numerical simulations and theoretical analysis demonstrated the performance of the developed jammer suppression method.

The following is the structure of the paper. The signal model of the QEPC–MIMO system is given in Section 2, while Section 3 provides the jamming suppression principles of the QEPC system. In Section 4, the CBNBC scheme was performed to enhance the robustness of mainlobe deceptive jamming suppression with direction-of-arrival (DOA) errors. To assess the performance of the aforementioned method, the results of numerical and simulation experiments are available in Section 5. Conclusions are drawn in Section 6.

## 2. Signal Model of QEPC-MIMO Radar

### 2.1. Quadratic Element Pulse Coding (QEPC) Scheme

To be generic, a collected MIMO radar with $M$ omnidirectional transmit elements and $N$ receive elements were considered in a uniform linear array as [4]. During a coherent interval (CPI), $K$ pulses were transmitted as shown in Figure 1. The $m$-th transmit element of the $K$-th pulse was expressed as

$$s_{m,k}(t) = \sqrt{\frac{E}{M}} c_{m,k} \phi_m(k) \mathrm{rect}\left(\frac{t}{T_p}\right) e^{j2\pi f_0(t+(k-1)T)} \tag{1}$$

where $E$ denoted the total energy, $T_p$ denoted the radar pulse duration. $f_0$ indicated the reference carrier. $\mathrm{rect}\left(\frac{t}{T_p}\right) = \begin{cases} 1, 0 < t < T_P \\ 0, else \end{cases}$, $\phi_m(k)$ indicated the complex envelope transmitted by the $m$-th element, which should be guaranteed as orthogonal waveform.

$$\phi_m(k) = \frac{1}{\sqrt{\tau_b}} \sum_{l=1}^{L} g_m(l) rect\left(\frac{t-(l-1)\tau_b}{\tau_b}\right), l = 1, \ldots, L \tag{2}$$

where $L$ and $\tau_b = \frac{T_p}{L}$ denoted the subpulse number and subpulse length, respectively, $g_m(l) = e^{jz_m(l)}, z_m(l) \in [0, 2\pi]$.

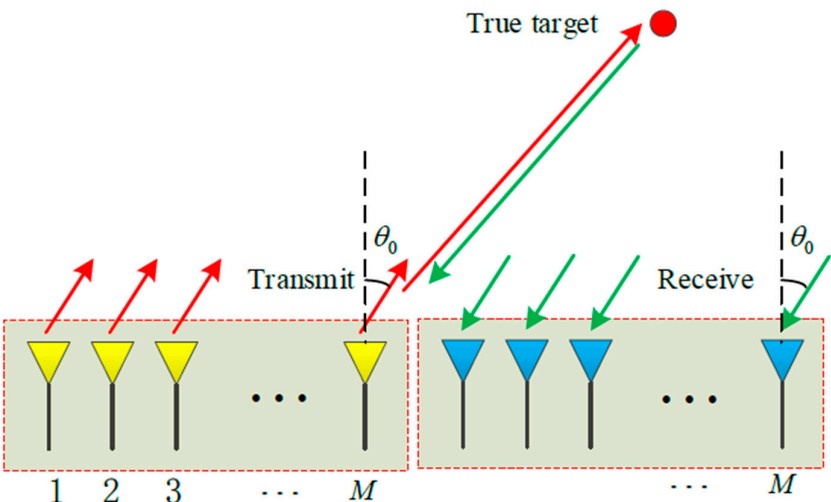

**Figure 1.** Signal model in the QEPC-MIMO.

The QEPC-MIMO radar had the initial phase, which was distinct from the conventional MIMO radar. $c_{m,k}$ indicated a special space-time code named QEPC factor. The QEPC factor of the $m$-th transmit spatial element and $k$-th temporal pulse could be represented as

$$c_{m,k}(\lambda_1, \lambda_2) = e^{j\pi\lambda_1(k-1+\lambda_2/\lambda_1(m-1))^2} \tag{3}$$

where $\lambda_1$ and $\lambda_2$ donated the fixed coding coefficients. Therefore, the QEPC vector of the $k$-th pulse was

$$c_k(\lambda_1, \lambda_2) = [c_{1,k}(\lambda_1, \lambda_2), c_{2,k}(\lambda_1, \lambda_2), \ldots, c_{M,k}(\lambda_1, \lambda_2)]^T \tag{4}$$

### 2.2. Receive Signal Model

Suppose a point-like target, whose range was $R_s$ and angle was $\theta_0$, located in the far-field (see Figure 1), where $0 < R_s < \frac{cT}{2}$ with $c$ represented light speed. Under the

narrowband assumption, the signal transmitted by $m$-th $(m = 1, \cdots, M)$ element and received by the $n$-th $(n = 1, \cdots, N)$ element was

$$x_{m,n,k}(t) = \partial_s \phi_{m,k}(t - \tau_0) c_{m,k}(\lambda_1, \lambda_2) e^{j2\pi f_0(t + (k-1)T_r - \tau_{m,n})} e^{j f_{ds}((k-1)T_r)} \tag{5}$$

where $\tau_{m,n} = \tau_0 - \frac{(m-1)d\sin(\theta_0)}{c} - \frac{(n-1)d\sin(\theta_0)}{c}$ indicated the round-trip propagation time delay. $\partial_s$ donated the complex coefficient of the point target, $d$ represented the inter-element space, $\tau_0 = \frac{2R}{c}$ was the common time delay. $f_{ds} = \frac{2v_s}{\lambda_0}$ was the Doppler frequency of the target with $\lambda_0$ and $v_s$, wavelength and the target speed, respectively.

As depicted in Figure 2, after multiplied by $e^{-j2\pi f_0 t}$, the measured signal were down converted. On each receive channel, the received waveform was decomposed into $M$ receive elements with $K$ pulses by a group of $M$ matched filters (MF) $h_l(t) = x_l^*(-t)$ $l = 1, \ldots, M$, which could be expressed as

$$\widehat{x}_{n,l,k}(t, \theta_0) = \beta_0 \phi_{l,k}(t - \tau_0) e^{j\pi\lambda_1(k-1+\lambda_2/\lambda_1(l-1))^2} e^{j2\pi\frac{d}{\lambda}(l-1)\sin(\theta_0)} e^{j2\pi\frac{d}{\lambda}(n-1)\sin(\theta_0)} e^{j2\pi f_{ds}(k-1)T_r} \sin c(t - \tau_0) \tag{6}$$

where $\beta_0 = \partial_s e^{-j2\pi f_0 \tau_0}$ represented the complex coefficient.

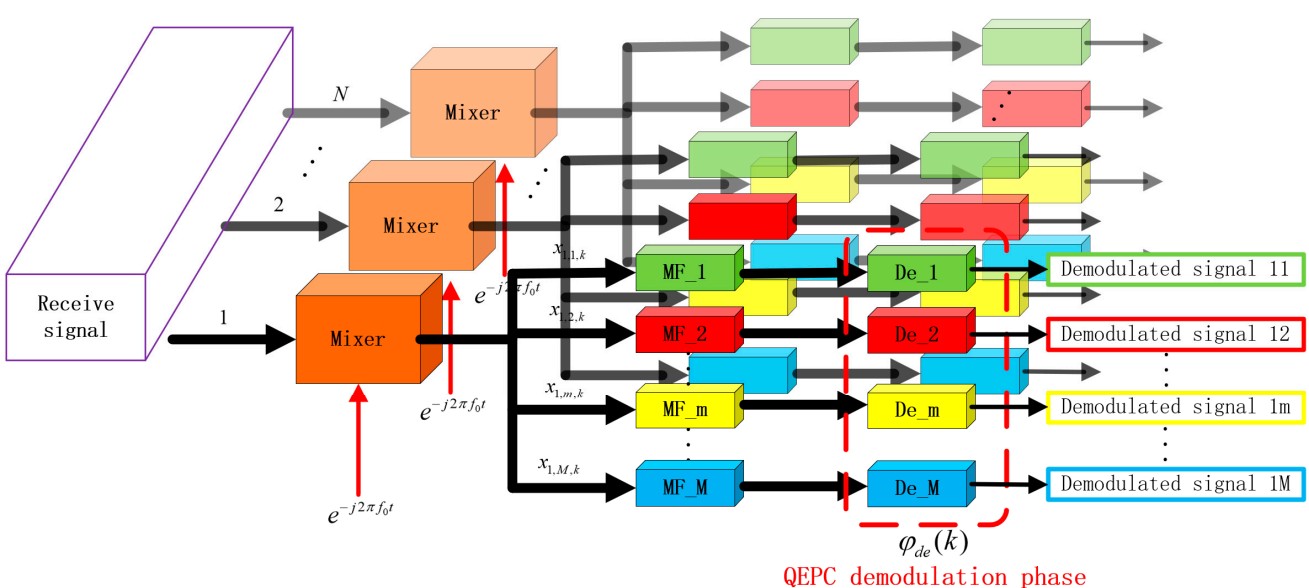

**Figure 2.** Receive signal processing procedures.

Assume the delay pulse number of the true target was $q_s$, and it arrived at the $k$-th pulse. The echo of the true target came from previous $k - q_s$th pulse and the true target owned the same code with the pulse, which was $c_{k-q_s}$. $MN$ transmit-receive pairs of the $k$-th pulse were stacked into an $MN \times K$-dimensional vector, which could be arranged as

$$\widehat{x}(t) = [\widehat{x}_{1,1,1}(t, \theta_0), \widehat{x}_{1,2,1}(t, \theta_0), \ldots, \widehat{x}_{n,m,1}(t, \theta_0), \ldots, \widehat{x}_{N,M,1}(t, \theta_0)]^T \otimes [1, e^{j2\pi f_{ds}T_r}, \ldots, e^{j2\pi f_{ds}(K-1)T_r}]$$
$$= \beta_0 r \odot [b(\theta_0) \otimes \widehat{a}(\lambda_1, \lambda_2, \theta_0)]^T \otimes h(f_{ds}) \tag{7}$$

where $r = \mathbf{1}_N \otimes \bar{r} \in C^{MN \times 1}$ with $\bar{r} = [\bar{r}_1, \bar{r}_2, \ldots, \bar{r}_M]^T \in C^{M \times 1}$ being the output after match filtering. $h(f_{ds})$, $a(t, \theta_0) \in C^{M \times 1}$, and $b(\theta_0) \in C^{N \times 1}$ represented the Doppler vector, the transmit steering vector, and the receive steering vector, respectively.

$$\widehat{a}(\lambda_1, \lambda_2, \theta_0) = [e^{j\pi(\lambda_1(k-q_s-1)^2)}, e^{j\pi(\lambda_1(k-q_s-1)^2 + 2\lambda_2(k-q_s-1) + \lambda_2^2/\lambda_1(m-1)^2)}, \ldots,$$
$$e^{j\pi(\lambda_1(k-q_s-1)^2 + 2\lambda_2(M-1)(k-q_s-1) + \lambda_2^2/\lambda_1(M-1)^2)}]^T \odot [1, e^{j2\pi\frac{d}{\lambda}\sin\theta}, \ldots, e^{j2\pi\frac{d}{\lambda}\sin\theta(M-1)}]^T \tag{8}$$
$$= c_k(\lambda_1, \lambda_2) \odot a(\theta_0)$$

$$h(f_{\text{ds}}) = [1, e^{j2\pi f_{\text{ds}}T_r}, \ldots, e^{j2\pi f_{\text{ds}}(K-1)T_r}] \odot [1, e^{-j2\pi q_s\lambda_1 T_r}, \ldots, e^{-j2\pi q_s\lambda_1(K-1)T_r}] \tag{9}$$

$$\boldsymbol{b}(\theta_0) = [1, e^{j2\pi\frac{d}{\lambda}\sin\theta_0}, \ldots, e^{j2\pi\frac{d}{\lambda}\sin\theta_0(N-1)}]^{\text{T}} \tag{10}$$

Accordingly, after the demodulation process being carried out, the decoding vector within the $k$-th pulse could be expressed as

$$\boldsymbol{g}_k = \mathbf{1}_N \otimes \boldsymbol{c}_{\text{de},k}(\lambda_1, \lambda_2) \tag{11}$$

where $\boldsymbol{c}_{\text{de},k}(\lambda_1, \lambda_2)$ and $\boldsymbol{c}_{\text{de},m,k}(\lambda_1, \lambda_2)$ represented the demodulation vector and demodulation factor, respectively. They could be expressed as

$$\boldsymbol{c}_{\text{de},k}(\lambda_1, \lambda_2) = [c_{\text{de},1,k}(\lambda_1, \lambda_2), c_{\text{de},2,k}(\lambda_1, \lambda_2), \ldots, c_{\text{de},M,k}(\lambda_1, \lambda_2)]^{\text{T}} \tag{12}$$

$$c_{\text{de},m,k}(\lambda_1, \lambda_2) = \exp\left\{-j\pi\lambda_1\left((k-1)^2 - 2\lambda_2(k-1)(m-1) - \lambda_2^2/\lambda_1^2(m-1)^2\right)\right\} \tag{13}$$

After demodulating pulse by pulse and element by element, the resident signal could be expressed as the form as

$$
\begin{aligned}
&\boldsymbol{y}_{\text{s},k}(t, \theta_0) \\
&= diag\{\boldsymbol{g}_k\}^\dagger \widehat{\boldsymbol{x}}_k(t, \theta_0) \\
&= \beta_0 e^{j2\pi f_{ds}(k-1)}(\mathbf{1}_N \otimes \boldsymbol{c}_{\text{de},k}^*(\lambda_1, \lambda_2)) \odot \boldsymbol{r} \odot [\boldsymbol{b}(\theta_0) \otimes \widehat{\boldsymbol{a}}^{\text{s}}(\lambda_1, \lambda_2, \theta_0)] \\
&= \beta_0 e^{j2\pi f_{ds}(k-1)}\boldsymbol{r} \odot \left\{(\mathbf{1}_N \odot \boldsymbol{b}(\theta_0)) \otimes [\boldsymbol{c}_{\text{de},k}^*(\lambda_1, \lambda_2) \odot \widehat{\boldsymbol{a}}^{\text{s}}(\lambda_1, \lambda_2, \theta_0)]\right\} \\
&= \beta_0 e^{j2\pi(f_{ds}-q_s\lambda_1)(k-1)}\boldsymbol{r} \odot \left\{\boldsymbol{b}(\theta_0)) \otimes \widetilde{\boldsymbol{a}}^{\text{s}}(\lambda_1, \lambda_2, \theta_0)\right\} \\
&= e^{-j\pi q_s^2\lambda_1}\beta_0 e^{j2\pi(f_{ds}-q_s\lambda_1)(k-1)}\boldsymbol{r} \odot \left\{\boldsymbol{b}(\theta_0)) \otimes \widetilde{\boldsymbol{a}}^{\text{s}}(\lambda_2, \theta_0)\right\} \\
&= \alpha_0 e^{j2\pi(f_{ds}-\Delta f_{\text{ds}})(k-1)}\boldsymbol{r} \odot \left\{\boldsymbol{b}(\theta_0)) \otimes \widetilde{\boldsymbol{a}}^{\text{s}}(\lambda_2, \theta_0)\right\}
\end{aligned}
\tag{14}
$$

where

- $\alpha_0 = e^{-j\pi q_s^2\lambda_1}\beta_0$ denoted complex echo coefficient of the true target.
- $\widetilde{\boldsymbol{a}}^{\text{s}}(\lambda_1, \lambda_2, \theta_0) = \boldsymbol{c}_{\text{de},k}^*(\lambda_1, \lambda_2) \odot \widehat{\boldsymbol{a}}^{\text{s}}(\lambda_1, \lambda_2, \theta_0)$
  $= e^{-j2\pi q_s\lambda_1(k-1)+j\pi q_s^2\lambda_1}[1, e^{-j2\pi q_s\lambda_2}, \ldots, e^{-j2\pi q_s\lambda_2(M-1)}]^{\text{T}}$
  $\odot [1, e^{j2\pi\frac{d}{\lambda}\sin\theta}, \ldots, e^{j2\pi\frac{d}{\lambda}\sin\theta(M-1)}]^{\text{T}} \in \mathbb{C}^M$ was the resident phase of the true target after demodulating.
- $\widetilde{\boldsymbol{a}}^{\text{s}}(\lambda_2, \theta_0) = [1, e^{-j2\pi q_s\lambda_2}, \ldots, e^{-j2\pi q_s\lambda_2(M-1)}]^{\text{T}} \odot [1, e^{j2\pi\frac{d}{\lambda}\sin\theta}, \ldots, e^{j2\pi\frac{d}{\lambda}\sin\theta(M-1)}]^{\text{T}}$ was the transmit steering vector of the true targets.
- $\boldsymbol{c}_{\text{de},k}^*(\lambda_1, \lambda_2) = e^{-j\pi\lambda_1((k-1)^2-2\lambda_2(k-1)(m-1))}$ denoted demodulation vector.
- $\Delta f_{\text{ds}} = q_s\lambda_1$ denoted the normalized Doppler shift frequency.

By stacking into a $MN \times K$ space-time snapshot, the signal was expressed as the form of

$$
\begin{aligned}
\boldsymbol{Y}_{\text{s}} &= [\boldsymbol{y}_{\text{s},1}, \boldsymbol{y}_{\text{s},2}, \ldots, \boldsymbol{y}_{\text{s},k}] \\
&= \alpha_s \boldsymbol{r} \odot \left\{\boldsymbol{b}(\theta_0)) \otimes \widetilde{\boldsymbol{a}}^q(\lambda_2, \theta_0)\right\} \otimes \boldsymbol{h}^{\text{s}}(f_{\text{ds}}, \lambda_1)
\end{aligned}
\tag{15}
$$

where $\boldsymbol{h}(f_{\text{ds}}, \lambda_1) = [1, e^{j2\pi(f_{\text{ds}}-q_j\lambda_1)}, \ldots, e^{j2\pi(f_{\text{ds}}-q_s\lambda_1)(K-1)}] \in \mathbb{C}^{1\times K}$ was the Doppler vector.

Considering the Gaussian noise in practice, the received signal could be expressed as

$$\boldsymbol{Y} = \boldsymbol{Y}_{\text{s}} + \boldsymbol{N} \tag{16}$$

where $\boldsymbol{N} \sim CN(0, \sigma^2 I_{MN})$ represented the white Gaussian noise, $\sigma^2$ and $I_{MN}$ donated the noise power and the $MN \times MN$-dimensional identity matrix, respectively.

## 3. Mainlobe Deceptive Jammer Suppression in the QEPC-MIMO Scheme

The principle of the mainlobe deceptive jammer suppression with the QEPC-MIMO radar was introduced. Firstly, the FTs that were generated and located in the Doppler-transmit-receive domain were investigated. Then, FTs and the true target were distinguished by the differences in the Doppler frequency as well as the transmit spatial frequency. In the end, by taking the distributions of targets into consideration, the jammers were suppressed in the joint Doppler-transmit-receive three-dimensional beamforming.

### 3.1. Generation of Mainlobe Deceptive Jammers

In order to fool the victim radar, the false target generator (FTG) intercepted the radar waveforms during the tracking phase and created a number of FTs which were pseudo-randomly distributed and had the proper temporal delayed time. Be aware that the FTs were typically located in the identical angle with the true target to ensure effective deception. As depicted in Figure 3, during the modulating step, the self-defense jammer was considered and FTG was situated at range $R_0$ and angle $\theta_0$ as [39]. In practice, the FTG usually delays the FTs to the next pulse(s), in order to generate FTs previous the true target in the fast-time dimension. We only focus in this situation.

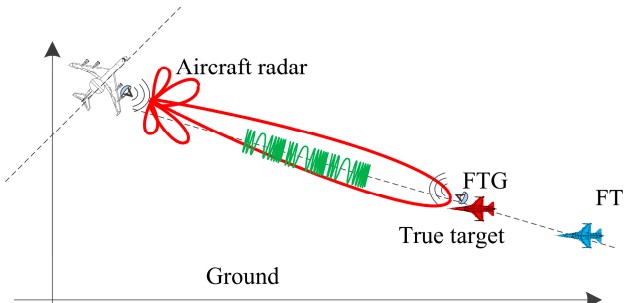

**Figure 3.** Scenario for airborne self-protection jamming.

As shown in Figure 4, based on the QEPC-MIMO radar, the targets were generated and located in the area. In this considered situation, False 1 and False 2 are located behind the true target at least one slow time pulse. FT 1 and FT 2 were generated from Pulse 2 and Pulse 1. In case 1, they were respectively located in the previous range bins, which were no overlap with the true target in the identical receive pulse. In case 2, they were located in the same range bin as the true target. All the cases could be suppressed in the latter section. Moreover, the QEPC was implemented among the array elements and the pulses.

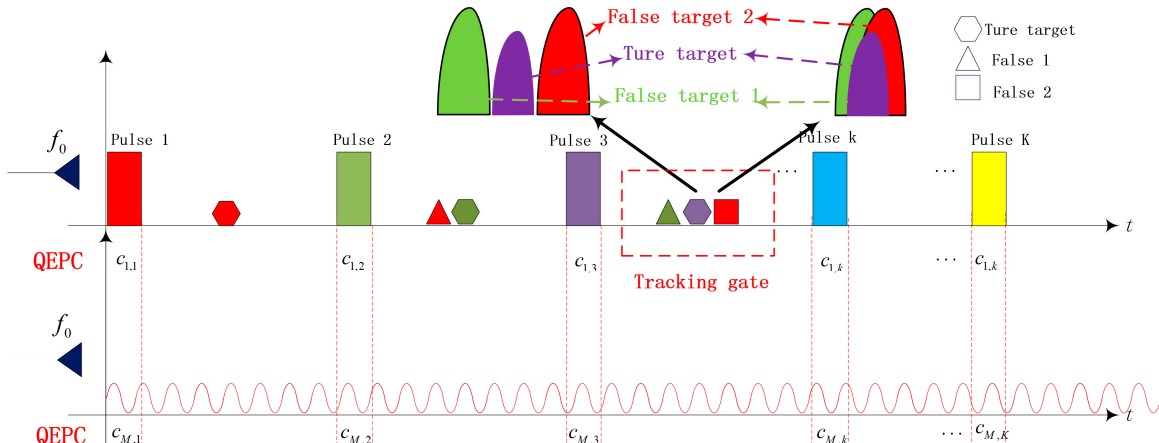

**Figure 4.** Illumination of true and FTs with the QEPC-MIMO.

There were $q$-th FT generated by the FTG, whose actual ranges were $R_q = \frac{c\Delta t_q}{2} + R_j$. After matched fliting, the signal of the $q$-th FT was written as

$$\tilde{x}^q(t) = \beta_q r \odot [b(\theta_0) \otimes a^q(\gamma_1, \theta_0)]^{\mathrm{T}} \otimes h(f_{\mathrm{d}q}) \tag{17}$$

After demodulating pulse by pulse and element by element, the resident signal could be expressed in the form as

$$
\begin{aligned}
&y_{q,k}(t, \theta_0) \\
&= diag\{g_k\}^\dagger \widehat{x}^q_{\ k}(t, \theta_0) \\
&= \beta_q e^{j2\pi f_{\mathrm{d}q}(k-1)} (1_N \otimes c^*_{\mathrm{de},k}(\lambda_1, \lambda_2)) \odot r \odot [b(\theta_0) \otimes \widehat{a}^q(\lambda_1, \lambda_2, \theta_0)] \\
&= \beta_q e^{j2\pi f_{\mathrm{d}q}(k-1)} r \odot \left\{ (1_N \odot b(\theta_0)) \otimes [c^*_{\mathrm{de},k}(\lambda_1, \lambda_2) \odot \widehat{a}^q(\lambda_1, \lambda_2, \theta_0)] \right\} \\
&= \beta_q e^{j2\pi (f_{dq} - q_q\lambda_1)(k-1)} r \odot \left\{ b(\theta_0)) \otimes \tilde{a}^q(\lambda_1, \lambda_2, \theta_0) \right\} \\
&= e^{-j\pi q_j^2 \lambda_1} \beta_q e^{j2\pi (f_{dq} - q_j\lambda_1)(k-1)} r \odot \left\{ b(\theta_0)) \otimes \tilde{a}^q(\lambda_2, \theta_0) \right\} \\
&= \alpha_q e^{j2\pi (f_{ds} - \Delta f_{ds})(k-1)} r \odot \left\{ b(\theta_0)) \otimes \tilde{a}^q(\lambda_2, \theta_0) \right\}
\end{aligned} \tag{18}
$$

where

- $\alpha_q = e^{-j\pi q_j^2 \lambda_1} \beta_q$ denoted complex echo coefficient of the $q$-th FT.
- $\tilde{a}^q(\lambda_1, \lambda_2, \theta_0) = c^*_{\mathrm{de},k}(\lambda_1, \lambda_2) \odot \widehat{a}^q(\lambda_1, \lambda_2, \theta_0)$
  $= e^{-j2\pi q_j\lambda_1(k-1) + j\pi q_j^2\lambda_1} [1, e^{-j2\pi q_j\lambda_2}, \ldots, e^{-j2\pi q_j\lambda_2(M-1)}]^{\mathrm{T}}$
  $\odot [1, e^{j2\pi \frac{d}{\lambda}\sin\theta}, \ldots, e^{j2\pi \frac{d}{\lambda}\sin\theta(M-1)}]^{\mathrm{T}} \in \mathbb{C}^M$ was the resident phase of the FT after demodulation.
- $\tilde{a}^q(\lambda_2, \theta_0) = [1, e^{-j2\pi q_j\lambda_2}, \ldots, e^{-j2\pi q_j\lambda_2(M-1)}]^{\mathrm{T}} \odot [1, e^{j2\pi \frac{d}{\lambda}\sin\theta}, \ldots, e^{j2\pi \frac{d}{\lambda}\sin\theta(M-1)}]^{\mathrm{T}}$ was the $q$-th FTs' transmit steering vector.
- $f_{dj} = \frac{2v_j}{\lambda_0}$ represented the normalized Doppler frequency of the FT, which was assumed to be the same as true target.
- $\Delta f_{dj} = q_j\lambda_1$ represented the normalized Doppler shift frequency of the FT.

By staking into an $MN \times K$ space-time snapshot, the signal was expressed in the form of

$$
\begin{aligned}
Y_q &= [y_{q,1}, y_{q,2}, \ldots, y_{q,k}] \\
&= \alpha_q r \odot \left\{ b(\theta_0)) \otimes \tilde{a}^q(\lambda_2, \theta_0) \right\} \otimes h^q(f_{\mathrm{d}q}, \lambda_1)
\end{aligned} \tag{19}
$$

where $h(f_{\mathrm{d}j}, \lambda_1) = [1, e^{j2\pi(f_{dj} - q_j\lambda_1)}, \ldots, e^{j2\pi(f_{dj} - q_j\lambda_1)(K-1)}] \in \mathbb{C}^{1 \times K}$ was the Doppler vector of the $q$-th FT.

Furthermore, considering the noise component and the true target, the total received signal was expressed as

$$Y = \sum_{q=1}^{Q} Y_q + Y_{\mathrm{sn}} = \sum_{q=1}^{Q} Y_q + Y_{\mathrm{s}} + N \tag{20}$$

### 3.2. Frequency Compensation and Distinguishment of the Targets

The transmit spatial frequencies of the true target and the $q$-th FT could be written as

$$f_{\mathrm{T,s}} = -\lambda_2 q_{\mathrm{s}} + \frac{d_{\mathrm{T}}}{\lambda_0} \sin(\theta_0) \tag{21}$$

$$f_{\mathrm{T,j}} = -\lambda_2 q_{\mathrm{j}} - \frac{d_{\mathrm{T}}}{\lambda_0} \sin(\theta_0) \tag{22}$$

The joint Doppler-transmit-receive compensating vector was constructed as

$$\boldsymbol{g} = \alpha_0 \mathbf{1}_N \otimes [1, e^{j2\pi q_s \lambda_2}, \ldots, e^{j2\pi q_s \lambda_2 (M-1)}]^{\mathrm{T}} \otimes [1, e^{j2\pi (q_s \lambda_1 - f_{ds})}, \ldots, e^{j2\pi (q_s \lambda_1 - f_{ds})(K-1)}] \quad (23)$$

Then the data was changed into the form as [37]

$$\tilde{\boldsymbol{Y}} = \boldsymbol{Y} \odot \boldsymbol{g} \quad (24)$$

After compensation, the transmit spatial steering vectors of the true target and the $q$-th FT were respectively written as

$$\hat{\boldsymbol{a}}^s (\lambda_2, \theta_0) = [1, e^{j2\pi \frac{d}{\lambda} \sin \theta_0}, \ldots, e^{j2\pi \frac{d}{\lambda} \sin \theta_0 (M-1)}]^{\mathrm{T}} \quad (25)$$

$$\hat{\boldsymbol{a}}^q (\lambda_2, \theta_0) = [1, e^{-j2\pi p \lambda_2}, \ldots, e^{-j2\pi p \lambda_2 (M-1)}]^{\mathrm{T}} \odot [1, e^{j2\pi \frac{d}{\lambda} \sin \theta}, \ldots, e^{j2\pi \frac{d}{\lambda} \sin \theta (M-1)}]^{\mathrm{T}} \quad (26)$$

After compensation, the normalized frequencies of the true target and the $q$-th FT could be written as

$$\hat{f}_{\mathrm{T,s}} = \frac{d_{\mathrm{T}}}{\lambda_0} \sin(\theta_0) \quad (27)$$

$$\hat{f}_{\mathrm{T,j}} = -\lambda_2 p + \frac{d_{\mathrm{T}}}{\lambda_0} \sin(\theta_0) \quad (28)$$

The difference in spatial frequency between the true target and the $q$-th FT could be given by

$$\Delta f_{\mathrm{T}} = \hat{f}_{\mathrm{T,s}} - \hat{f}_{\mathrm{T,j}} = \lambda_2 p \quad (29)$$

where $p = \left| q_s - q_j \right|$ was the difference in the delayed pulse number between the FTs and the true target.

After compensation, the Doppler vectors of the true target and the $q$-th FT were respectively written as

$$\hat{\boldsymbol{h}}^s (f_{\mathrm{d}s}, \lambda_1) = \mathbf{1}_{1 \times K} \in \mathrm{C}^{1 \times K} \quad (30)$$

$$\hat{\boldsymbol{h}}^q (f_{\mathrm{d}j}, \lambda_1) = [1, e^{-j2\pi p \lambda_1}, \ldots, e^{-j2\pi p \lambda_1 (K-1)}] \in \mathrm{C}^{1 \times \mathrm{K}} \quad (31)$$

The difference in normalized Doppler between the true and the FT was

$$\Delta f_{\mathrm{d}} = \Delta \hat{f}_{\mathrm{d}s} - \Delta \hat{f}_{\mathrm{d}q} = p \lambda_1 \quad (32)$$

From (29) and (32), by encoding, demodulation and compensation in the QEPC-MIMO radar, the location of the FT and the true target only depend on the delay pulse numbers in the spatial domain and Doppler domain. By the difference in the delay pulse numbers, the true target could be distinguished from the FTs in the joint Doppler-transmit-receive frequency domain.

### 3.3. Mainlobe Deceptive Jammer Suppression

To suppress these jammers, the data-independent beamformer was considered to apply. After compensation, the true target was located in the zero point in the joint Doppler-transmit-receive domain. The normalized equivalent transmit beampattern in transmitting spatial frequency domain could be given as

$$P_{\mathrm{T}}(f_{\mathrm{T}}) = \frac{1}{M} \frac{\sin(\pi M f_{\mathrm{T}})}{\sin(\pi f_{\mathrm{T}})} e^{j2\pi (M-1)(f_{\mathrm{T}})} \quad (33)$$

where $f_T$ represented the transmit spatial frequency. For the $q$-th FT, $f_T = f_T^q = \lambda_2 q_j + \frac{d_T}{\lambda_0}\sin(\theta) - \frac{d_T}{\lambda_0}\sin(\theta_0)$. In the equivalent transmit beampattern, it was desired that the FTs were aligned to the nulls while the true target was positioned at the mainlobe. To meet this requirement, the following conditions should be hold.

condition one: the denominator of $P_T\left(f_T^q\right)$ was not zero

$$f_T^q =\neq \frac{d}{\lambda_0} \cdot v \cdot \frac{\lambda_0}{d} = v v = 1, 2, \cdots, M - 1 \tag{34}$$

condition two: the numerator was zero

$$f_T^q = \frac{d}{\lambda_0} \cdot v \frac{\lambda_0}{Md} = v \cdot \frac{1}{M} \tag{35}$$

Meanwhile, the coding coefficient satisfied the followings
condition one:

$$\lambda_2 \neq \frac{v}{p} \tag{36}$$

condition two:

$$\lambda_2 = \frac{v}{pM} \tag{37}$$

In the QEPC-MIMO radar, the jammers were moved to the nulls by designing an appropriate $\lambda_2$. In the spatial domain, the maximum number of delayed pulses for repressible jammers was $M - 1$ due to $(M - 1)$ DOFs with the $M$-element array. That is, the FT whose delayed pulses number was $1 \sim M - 1$ could be suppressed by $1 \sim M - 1$-th nulls of the beampattern when $\lambda_2 = \frac{1}{M}$.

Similar to the suppression in the spatial frequency domain, the jammers could be distinguished and suppressed by designing an appropriate coding coefficient $\lambda_1$ in the Doppler domain. As stated in (32), the difference between the true one and the $q$-th FT was

$$\Delta f_d = p\lambda_1 = p(u + b) \tag{38}$$

where $b$ and $u$ were the fractional and integer part respectively. By reason of $2\pi$ periodicity, $u$ could be neglected. To distinguish the true target and the FTs, $p\lambda_1 \neq Z^+$ must be guaranteed. We assumed $b = \frac{1}{W}$, and then in the Doppler domain, the maximum number of delayed pulses for suppressible jammers was $W - 1$ due to $(W - 1)$ DOFs. The FT, whose delayed pulse number was $1 \sim W - 1$ could be respectively suppressed by $1 \sim M - 1$th null of the beampattern when $\lambda_1 = \frac{1}{W}$.

In Figure 5, the distribution of the targets in the joint transmit-Doppler domain was illustrated. The FT3 and the true one could not be distinguished for the identical spatial frequency. In the meanwhile, the true one and the FT2 could not be distinguished for identical Doppler frequency in the Doppler domain. However, with the QEPC-MIMO radar, all the FTs could be distinguished from the true target and suppressed via nulling in the joint Doppler-transmit-receive domain. In order to further enhance the ability to suppress the jammers, we designed $W$ and $M$ as prime, where the suppressible FTs number could reach the maximum, i.e., $WM - 1$.

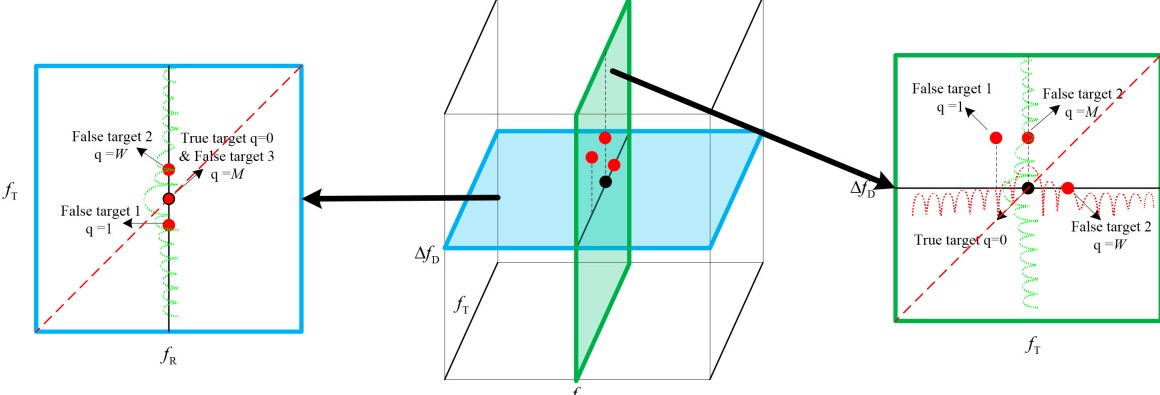

**Figure 5.** Distribution of the FTs and the true target in Doppler-transmit-receive space.

Moreover, to achieve the maximum output after compensation, the received signal was processed by the non-adaptive beamformer and the weight vector could be expressed as

$$\boldsymbol{w}_{QEPC} = \boldsymbol{b}(\theta_0)) \otimes \tilde{\boldsymbol{a}}^q(\lambda_1, \lambda_2, \theta_0) \tag{39}$$

The suppression method flow is shown in Figure 6.

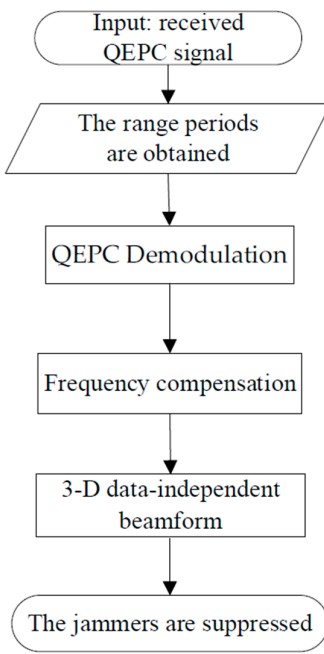

**Figure 6.** Processing flow chart.

## 4. Jammers Suppression with DOA Mismatch

### 4.1. Model Formulation

In practice, when quantization error and angular deviation occur, the spatial frequency deviations of the true and FTs are also present, which results in the mainlobe jammer suppression performance degrading. In such a case, the actual true target deviates from the center of the equivalent transmit beampattern and the actual FTs deviate from the presumed nulls. In this regard, with $\Delta\theta$ (value for the angular deviation), the true and FTs were represented as

$$\tilde{f}_{T,s} = \frac{d_T}{\lambda_0} \sin(\theta + \Delta\theta) \tag{40}$$

$$\tilde{f}_{\mathrm{T},j} = \lambda_2 p + \frac{d_\mathrm{T}}{\lambda_0} \sin(\theta + \Delta\theta) \tag{41}$$

However, the weight vector keeps the presumed value without deviation, which was $\boldsymbol{w}_{QEPC} = \boldsymbol{b}(\theta_0)) \otimes \tilde{\boldsymbol{a}}^q(\lambda_1, \lambda_2, \theta_0)$, resulting the performance of jammers suppression degrading and the power of the true target reducing. To solve the mentioned problem, we designed a data-independent 3-D beamformer with broadened deep nulls and flat-top mainlobe.

To adequately suppress the jammers, an adaptive theory was applicated in non-adaptive beamforming to design a Doppler-transmit-receive three-dimensional beampattern with wide nulls in the joint transmit-receive spatial frequency domain. That was, the weight vector $\boldsymbol{w}_*$ of this beamformer should satisfy the following conditions:

$$\min_{\boldsymbol{w}_*} \xi$$
$$\text{s.t.} \begin{cases} |\boldsymbol{w}_*{}^\mathrm{H}\boldsymbol{u}(f_\mathrm{T}, f_\mathrm{R})| \leq \xi, & (f_\mathrm{T}, f_\mathrm{R}) \in \Theta \\ \boldsymbol{w}_*{}^\mathrm{H}\boldsymbol{u}\left(f_\mathrm{T}^0, f_\mathrm{R}^0\right) = 1 \end{cases} \tag{42}$$

where $\Theta$ was the square region in which the jammers locate in, and it could be denoted as follows

$$\Theta \triangleq \left\{ (f_\mathrm{T}, f_\mathrm{R}) \quad \begin{array}{l} f_\mathrm{T} \in \left[f_\mathrm{T}^j - f_{\Delta\mathrm{T}}, f_\mathrm{T}^j + f_{\Delta\mathrm{T}}\right] \\ f_\mathrm{R} \in \left[f_\mathrm{R}^j - f_{\Delta\mathrm{R}}, f_\mathrm{R}^j + f_{\Delta\mathrm{R}}\right] \end{array} \right\} \tag{43}$$

where $f_\mathrm{R}^j$ and $f_\mathrm{T}^j$ respectively represented the receive frequency and the transmit frequency of the theoretical jammers. $f_{\Delta\mathrm{R}}$ and $f_{\Delta\mathrm{T}}$ respectively represented the maximum allowable deviations of the receive frequency and the transmit frequency, and $\xi$ represented the depth of the predefined nulls for the beampattern.

### 4.2. Null Broadening Adaptive Beamforming Formulation

In the section, the CBNBC was proposed to broaden nulls in the transmit-receive beampattern. As depicted in Figure 7, the square $\Theta$ which the FTs with spatial frequency deviation might be located in, were constructed in the spatial frequency domain. Lots of artificial jammers were generated via $l(l = 1, 2, \cdots)$ iterations. Then, the receive and transmit spatial frequencies of the $l$th artificial jammers were respectively written as

$$f_{\mathrm{R}\,j}^l = \begin{cases} f_\mathrm{R}^j + lf_\Delta, & l = 4\updownarrow + 1 \\ f_\mathrm{R}^j - lf_\Delta, & l = 4\updownarrow + 2 \\ f_\mathrm{R}^j - lf_\Delta, & l = 4\updownarrow + 3 \\ f_\mathrm{R}^j + lf_\Delta, & l = 4\updownarrow + 4 \end{cases} \quad f_{\mathrm{T}\,j}^l = \begin{cases} f_\mathrm{T}^j + lf_\Delta, & l = 4\updownarrow + 1 \\ f_\mathrm{T}^j + lf_\Delta, & l = 4\updownarrow + 2 \\ f_\mathrm{T}^j - lf_\Delta, & l = 4\updownarrow + 3 \\ f_\mathrm{T}^j - lf_\Delta, & l = 4\updownarrow + 4 \end{cases} \tag{44}$$

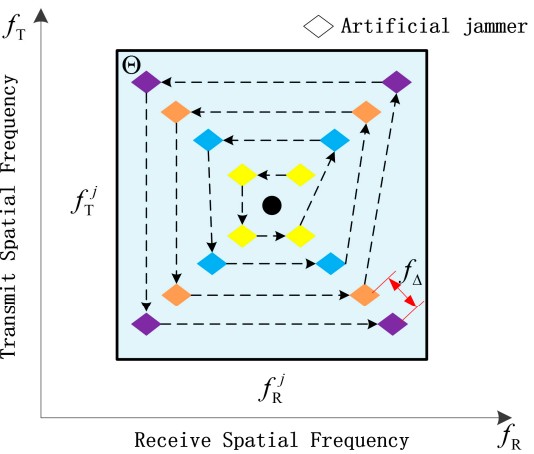

**Figure 7.** The square null region with artificial jammers.

Composing jammers by *l*-th iteration, the jammer-plus-noise covariance matrix could be constructed as

$$R_{j+n} = R^{\{l\}} = R^{\{l-1\}} + \sigma_l^2 u_l u_l^\dagger \tag{45}$$

where $u_l \triangleq u(f_T, f_R)$ represented the jammers steering vector of the *l*-th iteration, $R^{\{0\}}$ was the noise matrix and $\sigma_l^2$ represented the power of the jammers by *l*-th iteration. $(\cdot)^\dagger$ denotes conjugate transpose operators.

From (39), the response at $u_l$ of the beampattern could be expressed as

$$P(u_l | u_0) = w_*^{\mathrm{H}} u(f_T, f_R) = \xi \tag{46}$$

Hence, according to the MVDR criterion and CBNBC, the weight vector could be calculated as

$$w_l = \Lambda_l \left( R^{\{l\}} \right)^{-1} u_0 = \Lambda_l \left[ -\frac{u_l^{\mathrm{H}} \left( R^{\{l-1\}} \right)^{-1} u_0 \sigma_l^2 \left( R^{\{l-1\}} \right)^{-1} u_l}{1 + \sigma_l^2 u_l^{\mathrm{H}} \left( R^{\{l-1\}} \right)^{-1} u_l} + \left( R^{\{l-1\}} \right)^{-1} u_0 \right] \tag{47}$$

where $\Lambda_l = \left( u_0^{\mathrm{H}} \left( R^{\{l\}} \right)^{-1} u_0 \right)^{-1}$

The power of the *l*th artificial jammer was calculated as

$$\sigma_l^2 = \frac{u_0^{\mathrm{H}} (u_l - \xi_l u_0) \left( R^{\{l-1\}} \right)^{-1}}{\xi \left[ u_0 u_l^{\mathrm{H}} \left( R^{\{l-1\}} \right)^{-1} u_l - u_l u_l^{\mathrm{H}} \left( R^{\{l-1\}} \right)^{-1} u_0 \right] u_0^{\mathrm{H}} \left( R^{\{l-1\}} \right)^{-1}} \tag{48}$$

Moreover, $\theta_h$ ($h = 1, 2, \ldots, H$) was the value that approximates the sidelobe situations to control the sidelobe level, which was written as the following:

$$\left| w_l^{\mathrm{H}} u(\theta_h) \right| < \delta_h \tag{49}$$

Define a difference function to represent the difference between the response and the expected value of the beampattern in region $\Theta$.

$$D_{sl} = \int_{f \in \Theta} \min \left( \xi_l - \left| w_l^{\mathrm{H}} u(f) \right|, 0 \right) df \tag{50}$$

where $u(f)$ represented all the preset jammers' steering vector. The iteration stopped when the $D_{sl} \le \varepsilon$, which represented the symbol to stop iterating.

## 5. Simulations

### 5.1. Perfomance of Ideal Jammer Suppression

In this section, numerical simulations and theoretical analysis were presented to assess the performance of jammers suppression method and the robustness against jammers in the presence of deviations with the QEPC-MIMO radar. As listed in Table 1, the simulation parameters of the radar system are provided. Consider a collocated MIMO radar with 15 collocated transmit-receive antennas, operating in 15 GHz with pulse repetition frequency at 10 kHz. To be generic, the inter-element spaces are designed as half a wavelength at 0.0093 m. In Table 2, the direction and position of the true target are 0° and 9 km, respectively, which has no range ambiguity for simplicity. For FTs, which have the same angle as the true target, are generated behind the true one for 1, 1, 15, and 16 delayed pulses, which are located in 200, 350, 150 and 400, respectively.

**Table 1.** Parameters of the QEPC-MIMO system.

| Parameter | Value | Parameter | Value |
|---|---|---|---|
| Transmit elements space | 0.0093 m | Receive elements space | 0.0093 m |
| Transmit elements number | 15 | Receive elements number | 15 |
| The carrier frequency | 15 GHz | Pulse repetition frequency | 10 kHz |
| Coding coefficient $\lambda_2$ | 1/15 | Coding coefficient $\lambda_1$ | 1/16 |

**Table 2.** Parameters of targets.

| | True Target | FT 1 | FT 2 | FT 3 | FT 4 |
|---|---|---|---|---|---|
| Angle (°) | 0 | 0 | 0 | 0 | 0 |
| Range(km) | 9 | 4.5 | 6 | 10.5 | 12 |
| Range bin | 300 | 150 | 200 | 350 | 400 |
| Time delay (ms) | 0 | 1.47 | 0.08 | 0.11 | 1.62 |
| Velocity (m/s) | 20 | 20 | 20 | 20 | 20 |
| SNR (dB) | 20 | \ | \ | \ | \ |
| JNR (dB) | \ | 20 | 20 | 25 | 25 |
| Delayed pulse | 0 | 15 | 1 | 1 | 16 |

Figure 8 intuitively visible illustrates the spectrum Distributions of targets. The Capon in the spatial domain was illustrated in Figure 8a,b. With a uniform receive spatial frequency, the targets occupy a straight line. The transmit spatial frequencies vary based on the number of delayed pulses. That was to say, the targets concentrated at the same transmit spatial frequency when their delayed pulses were identical. Consequently, FT1 remains indistinguishable from the true target due to its identical position in targets' range-Doppler spectrum Distributions. Figure 8c displays the targets' range-Doppler spectrum Distributions. The true target and FTs exhibit two distinct peaks when they were projected onto the Doppler domain. Based on the number of delayed pulses, the normalized Doppler frequency frequencies vary. True target differs from the FTs except FT4, which has the identical normalized Doppler frequencies with the true one. Therefore, in order to discriminate all FTs, especially FT1 and FT4, from the true one, the joint Doppler-transmit-receive domain was considered.

As shown in Figure 9a, the 3D data-independent beamforming with the QEPC-MIMO radar is presented in the joint Doppler-transmit-receive domain. Figure 8b displays the slice of the 3D beampattern, which illustrates the transmit-receive beampattern. Processed by the compensating vector, the true one is moved to the center of the 3-D beampattern. The dots representing the FTs are projected in Figure 9b, where the FDA-MIMO or EPC-MIMO have the identical transmit-receive beampattern. Traditional FDA and EPC radar systems have difficulty in differentiating between the true target and FTs (such as FT1), which share the same spatial frequency with the true one. However, with the use of the QEPC-MIMO radar technology, it is possible to suppress all FTs in the joint Doppler-transmit-receive domain.

Furthermore, Figure 10a provides a comparison of output powers among different radar systems. Based on the QEPC-MIMO radar, the FTs are suppressed by nulling in the joint Doppler-transmit-receive domain and then the true one exhibits a superior output power level. Conversely, the FT (such as FT1), which has $M$ delayed pulses, cannot be effectively suppressed by the FDA-MIMO or the EPC-MIMO radar systems. Figure 10b provides a comparison of output powers among different radar systems when multiple true targets exist. Similarly, the jammers in multi-objective scenarios can also be suppressed. As shown in Figure 10c, the signal-to-jammer-plus-noise ratio (SJNR) curves of different radar frameworks are displayed as a function of input SNR with 300 Monte Carlo (MC) trials. An ideal case is presented as the upper bound. It is noteworthy that the FTs with delayed pulse

numbers of $M$ cannot be suppressed in the FDA-MIMO radar and the EPC-MIMO radar, resulting in output degrading. Contrary to this, the output SJNR performance improves in the QEPC-MIMO radar by effectively nulling jammers based on the data-independent Doppler-transmit-receive 3D beamforming.

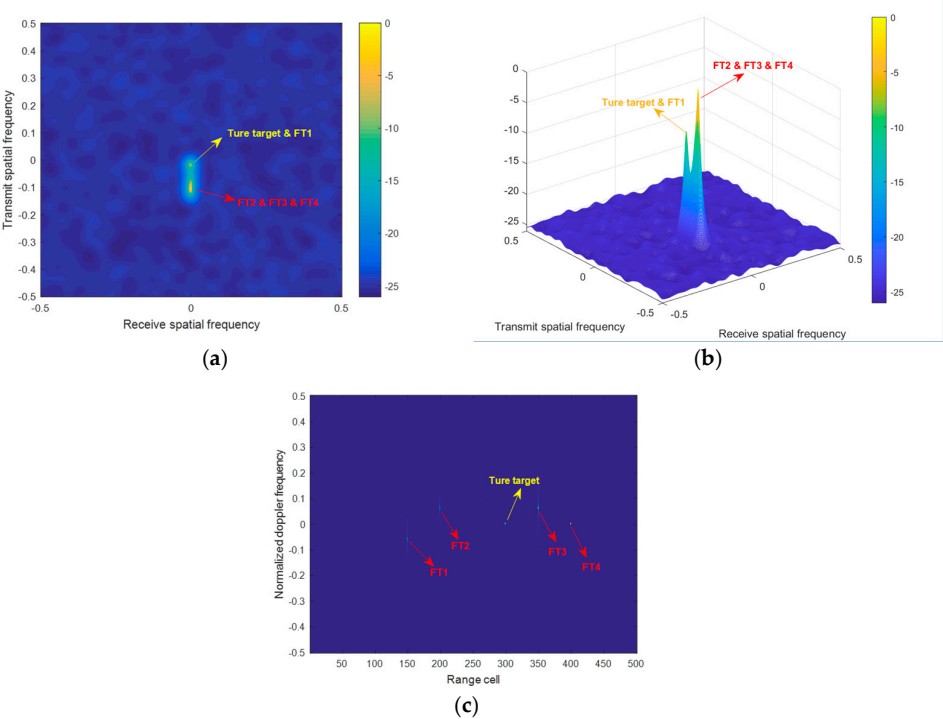

**Figure 8.** Spectrum Distributions (**a**) Capon in the spatial domain. (**b**) 3-D plot in the Transmit–Receive domain. (**c**) 2-D plot in Range-Doppler domain.

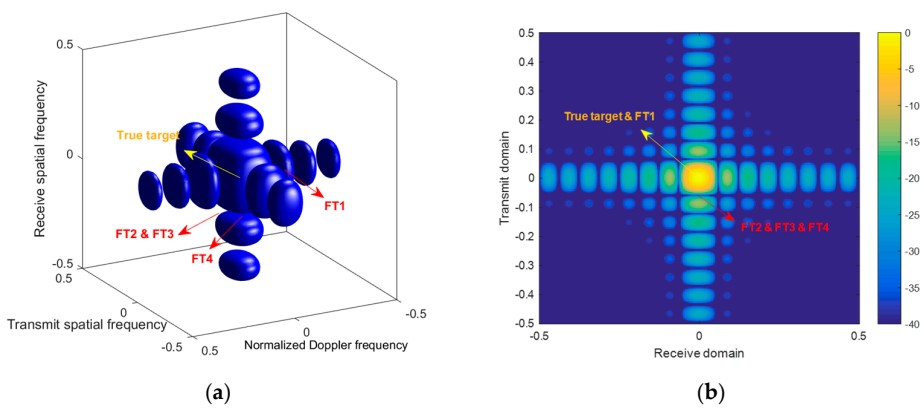

**Figure 9.** Beampatterns. (**a**) Doppler-transmit-receive 3D beampattern in the QEPC-MIMO radar. (**b**) Transmit-receive slice of the 3D beampattern in the QEPC-MIMO radar.

Figure 11 plots the nulls pointing towards FTs in different radar frameworks. The coding coefficient is denoted on the y-axis and the delayed pulse number is displayed on the x-axis. Particularly, when the suppression is effective, the yellow color exists and when the suppression is invalid, the blue color exists. As described in Figure 11a,b, when the frequency increment in FDA-MIMO and coding coefficient in EPC-MIMO are given as $\gamma_1 = \gamma_2 = 1/M$, the suppressible jammer maximum number is $M - 1$. The suppression becomes invalid to jammers with identical transmission frequency as the true target (the exponential term has a period of $2\pi$). As described in Figure 11c,d, when $\lambda_2 = 1/M$ and $\lambda_1 = 1/W$ are designated to be coprime, the suppressible jammers' maximum number is

$(M-1)(W-1)$. Consequently, there are significant advantages to maximize the suppressible jammers number by utilizing mentioned suppression method, which aligns with the theoretical analyses.

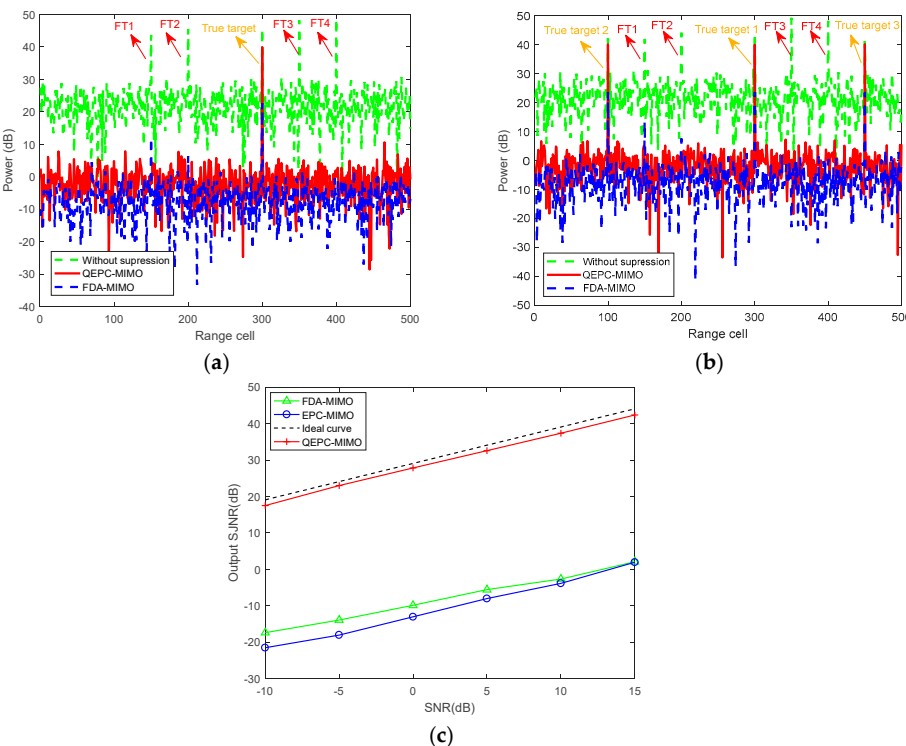

**Figure 10.** Comparison of output results. (**a**) Comparison in one true target (**b**) Comparison in multiple true targets (**c**) Output SJNR performance with respect to input SNR.

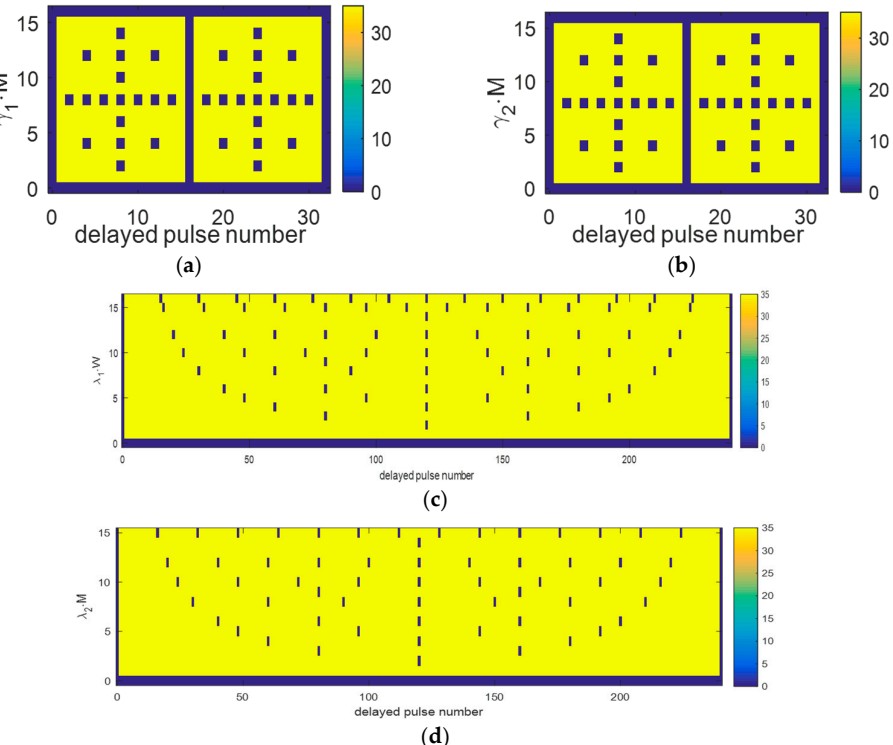

**Figure 11.** Suppression nulls of FTs. (**a**) FDA-MIMO. (**b**) EPC-MIMO. (**c**) $\lambda_1$ versus nulls number in the QEPC-MIMO radar. (**d**) $\lambda_2$ versus nulls number in the QEPC-MIMO radar.

### 5.2. Performance of Robust Jammer Suppression

In this section, the CBNBC method is verified by the results of numerical and simulation experiments in Table 3. The angle estimation errors for FT1, FT2, FT3, and FT4 are defined as 1.5, 2, 2, and 1.5, respectively. In the meanwhile, the delayed pulses for FT1, FT2, FT3, and FT4 are defined as 1, 1, 15, and 16, respectively. Table 3 displays the detailed parameters.

**Table 3.** Parameters of targets with deviations.

|  | True Target | FT 1 | FT 2 | FT 3 | FT 4 |
|---|---|---|---|---|---|
| Angle error(°) | 1 | 1.5 | 2 | 2 | 1.5 |
| Angle (°) | 1 | −1.5 | 2 | −2 | 1.5 |
| Delayed pulse | 0 | 1 | 1 | 15 | 16 |
| Range (km) | 9.01 | 4.52 | 5.98 | 10.515 | 11.985 |
| Range bin | 300 | 170 | 200 | 255 | 400 |
| Time delay (ms) | 0 | 0.074066 | 0.079800 | 1.490033 | 1.619835 |
| SNR (dB) | 20 | \ | \ | \ | \ |
| JNR (dB) | \ | 20 | 20 | 25 | 25 |

Figure 12 intuitively visible illustrates the spectrum distributions of artificial FTs based on the CBNBC-QEPC-MIMO radar. After applying the CBNBC method, there are the artificial FTs generated in a square-like region. According to (44), the artificial FTs are arranged around the presumed FT, ensuring that the actual FTs with deviations involve in the region. Then, the noise and jammers covariance matrix are improved in (45), allowing the formed weight vector to suppress the jammers with deviations.

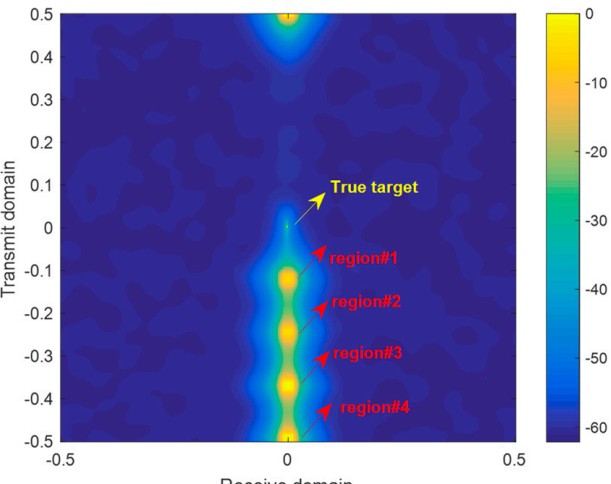

**Figure 12.** Capon in Spectrum Distributions based on the CBNBC-QEPC-MIMO radar.

Figure 13 intuitively visible illustrates the beampattern in CBNBC-QEPC-MIMO radar. The transmit-receive 2-D beampattern, where four broadened notches are predefined, is obtained in Figure 13a. The FTs with quantization error and angle estimation error dwell in the broadened notches generated by CBNBC. From the cross-sectional image formed by the red line in the transmit-receive 2-D beampattern, the equivalent transmit beampattern is obtained in Figure 13b. Based on CBNBC method, the broadened deep nulls and flat-top mainlobe can be accomplished, which has superiority in suppressing the FTs with deviations.

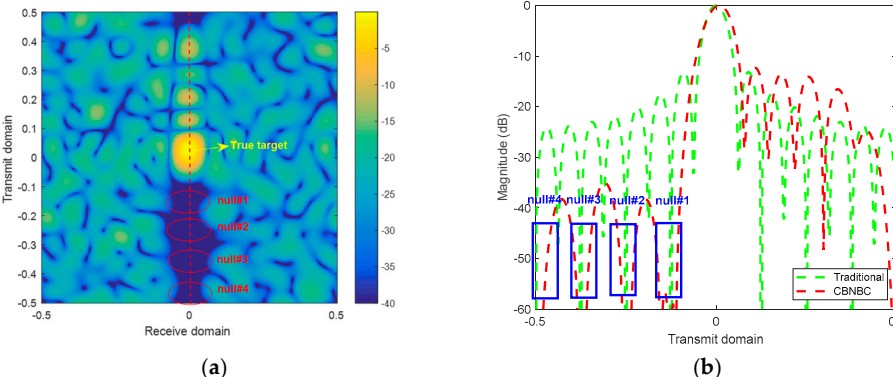

**Figure 13.** Beampatterns with CBNBC method. (**a**) Transmit-receive beampattern. (**b**) Equivalent transmit beampattern.

As is depicted in Figure 14, the comparisons of output power with CBNBC beamformer and original beamformers are obtained. The mentioned methods can effortlessly suppress FTs which are situated in predefined regions. Nevertheless, FTs cannot be suppressed by the original beamformer for the reason that the notches could not align to the FTs. Hence, the method proposed in this paper has superiority in effectively suppressing the jammers with quantization and angular deviations. It is consistent with the theoretical analyses.

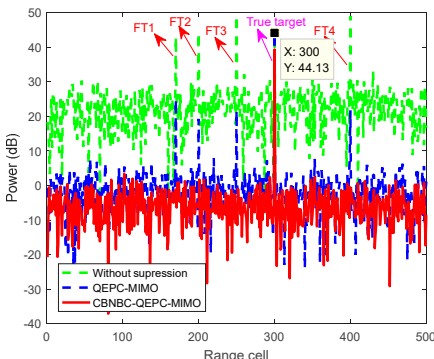

**Figure 14.** Output results of CBNBC.

## 6. Discussion

The traditional FDA-MIMO and EPC-MIMO radar systems have difficulty in distinguishing the targets for the limitation of DOFs. However, by applying the QEPC in transmitted pulses of different elements in MIMO radar, the true target differs from the FTs in the joint Doppler-transmit-receive domain for additional DOFs in Figures 8 and 9. As shown in Figure 10a,b, it is noteworthy that the FTs with delayed pulse numbers of $M$ cannot be suppressed in FDA-MIMO radar and EPC-MIMO radar, resulting in the output degrading in Figure 10c. Contrary to this, the output SJNR performance has been improved in the QEPC-MIMO radar by effectively nulling jammers in the joint Doppler and spatial frequency domain. As shown in Figure 11, the maximum numbers of the suppressible jammer in FDA-MIMO and in EPC-MIMO are $M-1$. However, when $\lambda_2 = 1/M$ and $\lambda_1 = 1/W$ are designated to be coprime, the maximum number of the suppressible jammers is $(M-1)(W-1)$. Consequently, there are significant advantages to maximize the suppressible jammers number by utilizing mentioned suppression method.

To suppress the false targets with deviations, the CBNBC is performed. The noise and jammers covariance matrix are improved in (45), allowing the broadened deep nulls and flat-top mainlobe in Figure 12. FTs cannot be suppressed by the original beamformer for the reason that the notches could not accurately align to the FTs in Figure 13. Hence, the capability to suppress jammers with deviations can be strengthened.



## 7. Conclusions

In this paper, in order to suppress the mainlobe deceptive jammer, the QEPC-MIMO scheme has been developed. In the transmit array, the QEPC has been performed by coding the transmitted pulses of different spatial channels. In the receiver, the Doppler frequency and the spatial frequency offsets have been utilized to move the jammers to sidelobes in the joint Doppler-transmit-receive domain. Then, further spatial frequency compensation and Doppler compensation have been performed to align the true target to the zero point in the transmit-spatial and Doppler domains. Moreover, by designing appropriate coding coefficients, the jammers have been suppressed via data-independent Doppler-transmit-receive three-dimensional beamforming. The CBNBC approach has been performed to enhance the robustness against angular estimation mismatches, which has broadened the nulls in the equivalent beampattern by generating multiple artificial jammers with preset powers around the nulls. Hence, the actual FTs with deviations have been sufficiently suppressed in the broadened notches. Numerical simulations and theoretical analysis have highlighted the performance of the developed jammer suppression method with quantization and angle estimation errors.

In the future, the research could concentrate on exploring the suppression of mainlobe deceptive jammers by utilizing real radar data with physical array while accounting for the existence of clutter.

**Author Contributions:** Conceptualization, Y.Z. and J.X.; methodology, Y.Z. and J.X.; software, L.L.; validation, Y.Z., J.X., G.L. and L.L.; formal analysis, Y.Z. and L.L.; investigation, Y.Z., L.L. and J.X.; resources, G.L.; data curation, Y.Z. and J.X.; writing—original draft preparation, Y.Z.; writing—review and editing, Y.Z., J.X. and L.L.; visualization, Y.Z. and L.L.; supervision, L.L., G.L. and J.X.; project administration, X.Z., L.L., G.L. and J.X.; funding acquisition, X.Z., L.L., G.L. and J.X. All authors have read and agreed to the published version of the manuscript.

**Funding:** This work was supported in part by the National Nature Science Foundation of China (Nos. 62101402, 61931016, 62071344), China Postdoctoral Science Foundation (Nos. 2021TQ0261, 2021M702547); SAST Innovation fund (No. SAST2022-041), Natural Science Basic Research Program of Shaanxi (No. 2023-JC-JQ-55), Innovation Capability Support Program of Shaanxi (No. 2022KJXX-38).

**Data Availability Statement:** Not applicable.

**Conflicts of Interest:** The authors declare no conflict of interest.

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
