# Peer review of "Suppression of Mainlobe Jammers with Quadratic Element Pulse Coding in MIMO Radar"

_remotesensing, doi:10.3390/rs15123202_

Round 1

Reviewer 1 Report

Comments on remotesensing-2391425
1. T
his paper exhibits conciseness, clear writing, and a well-organized structure. From
my perspective, the utilization of QEPC in MIMO increases the maximum number of
suppressible jammers and the application of CBNBC enhance the performance to
suppress the mainlobe deceptive jammers with angular estimation deviations. The
author can clarify the point more clearly.

2. What is the prior information for interference suppression? Can all the mainlobe
deceptive interferences be suppressed? What are the limitations involved? It is
recommended that the author can provide a clearer description.

3. The formulaic symbols within the article should be thoroughly reviewed to ensure
accuracy. Additionally, the grammar should be further refined to align with
professional standards. In formula (17),
the letter ‘d’ indqf is straight, however, the
letter ‘d’ in
dqf of formula (18) is Italic.
4. In this article, the clarity of the schematic diagram is inadequate (such as Figure 4). It
is recommended to consider either altering the image format or generating a new
diagram to enhance its visual comprehensibility.

5.
In the section of simulation results, the titles of simulations should follow a consistent
format, rather than both ‘
(a)’ and ‘(a)’ coexisting.
6
. In formula (45), the symbol ’ should be explained in your paper to understand it
clearl

Reviewer 2 Report

Thank the authors for preparing the work for publication in this journal. In this work, the authors consider an effective suppression method with QEPC along the slow time in FDA-MIMO radar, which can be applied for mainlobe deceptive jammer suppression problem with angular estimation deviations. As I reviewed the paper, my comments are attached.

Point 1: In conclusion, the paper is concise, well written and organized. The advantages of the method in my opinion are two point: 1.By applying a novel code in MIMO, limitation of the maximum number of suppressible jammers are improved; 2. The mainlobe deceptive jammer with angular estimation deviations can be suppressed via CBNBC in QEPC-MIMO radar. The ideas are sounded and interesting. However, the advantage can be explained more clearly and detailedly.

Point 2: More representative literature should be cited in references. Compared to EPC and FDA, which aspects the advantages of the mentioned method are reflected in? The authors should state clearly the contribution of this work. The citation should be correctly used to avoid redundancy.

Point 3: In this paper, by utilizing the additional Doppler and spatial frequency offset generated via decoding, the jammers are equivalently escaped from the mainlobe in the transmit-receive-Doppler domain. Can you provide more details about additional Doppler and spatial frequency offset and how false targets migration is achieved?

Point 4: On page 4, there are some mistakes located in M matched filters of Figure 2, please correct it and make it clearer.

Point 5: This work's symbols should be carefully checked by the authors, and the English usage should be proofread by a professional.

Point 6: In the article, the colors of some targets used in the simulation figures could be replaced to make the images clearer.

The use of English is satisfactory.

Reviewer 3 Report

It would be better if you observed the common IMRAD structure of the manuscript. If possible, please, restructure your paper with accordance to IMRAD.

Are Figures 1, 2, 3, etc. your own? If some of them are not the product of your own research and graphical work, there should be a reference to the source. The same remark is true for all the formula you used in the study. If they are not a product of your own work, there should be a quotation for each equation.

Figure 4 is somewhat difficult to comprehend. If possible, please, rebuild it to make it more clear.

Yellow color for the caption of True target in the Figure 7b looks exremely difficult to read. Please, change the color scheme for the text.

Figures 10a and 10b look identically. Why?

In general, the work is really interesting and prospective. 

Reviewer 4 Report

Authors present a method for suppressing mainlobe jammers with quadratic element pulse coding in MIMO radar. There exists 6 major issues and 1 minor issues. Comments are listed as follows.

Major issue 1: Authors should strengthen the description of novelty and contribution of this manuscript in the part of introduction.

Major issue 2: Authors should present a flowchart which contains steps of the proposed method.

Major issue 3:Authors should also present numerical analysis of the results of simulation.

Major issue 4:If it is possible, authors should use real radar data to validate the proposed method.

Major issue 5: As presented in Figure 12, jammers are suppressed, but sidelobe in right side becomes much higher. This problem can not be ignored. Authors should improve the proposed method and reduce this phenomenon.

Major issue 6: Only one point target scene in presented in this manuscript. Authors should also present simulation with multiple true targets and jammers in the part of experiment.

Minor issue 1: In Table 3, the SNR and JNR should be separated in different rows.

Minor editing of English language required

Round 2

Reviewer 4 Report

I think this paper has been revised carefully according to the comments from reviewers. This manuscript can be accepted for publication in this version.